# Towards a Better Workplace Environment—Empirical Measurement to Manage Diversity in the Workplace

**DOI:** 10.3390/ijerph192315851

**Published:** 2022-11-28

**Authors:** Elwira Gross-Gołacka, Teresa Kupczyk, Justyna Wiktorowicz

**Affiliations:** 1Department of Organization Theory and Methods, Faculty of Management, University of Warsaw, 02-678 Warsaw, Poland; 2Department of Management, General Tadeusz Kościuszko Military University of Land Forces, 51-147 Wroclaw, Poland; 3Department of Economic and Social Statistics, Faculty of Economics and Sociology, University of Lodz, 90-214 Lodz, Poland

**Keywords:** workplace environment, diverse workforce, diversity management, measurement

## Abstract

(1) Background: In this article, we explore the ever-present problem of achieving better results in the area of creating innovative and diverse human resources in the workplace environment through learning and inference from past actions. (2) Methods: An original proposal of five synthetic indicators was developed, corresponding to individual areas of the 5P architecture. In order to test the homogeneity of the index, exploratory factor analysis was used. The reliability of the new indicator was also assessed, both in total and in selected sub-areas. The value of the synthetic index was determined as a summary score in the selected areas (the sum of the values of individual diagnostic variables). Finally, the distribution of synthetic indicators WP1–WP5 was analyzed. (3) Results: By assumption, this analysis was performed to enable an empirical verification of the theoretical model, which combined the strategic, tactical and operational levels, delineating five steps (areas) that should be taken to create an organization open to diversity and its management. The presented approach also allowed for the visualization of the implementation of the organization’s activities in individual areas of diversity management expressed in the proprietary 5P architecture in many dimensions (planning, implementation and monitoring). In addition, the analyses confirmed that the individual dimensions of the diversity management architecture interacted with each other and that the direction of this correlation was positive: the development of diversity management in different areas occurred in parallel, but nevertheless, as can be seen from the values of the correlation coefficients, at a different pace/range. (4) Conclusions: The use of quantitative methods in the decision-making process of an organization can have a significant impact on the quality of its management. In the case of building an inclusive environment and implementing activities for diversity management, the proposed 5P architecture could significantly support this process. Therefore, it is recommended to use the proposed 5P architecture in practice, for example, to diagnose the scope and quality of actions taken for diversity management, as well as to build a diverse working environment in key areas of the organization.

## 1. Introduction

In today’s dynamic reality, managers are working to improve their workplace environment, organization and its products or services. They are looking for solutions that will allow them to get to the top of the market or to maintain their position in a world overwhelmed by the need to implement innovations. Every day, we see countless new models and innovative solutions emerge, some of which are forgotten or dismissed. This particular phenomenon of diversity of human resources is nothing new, it has always been present in societies, which means that it has always been present in organizations as well. People are different, no matter how similar they are. The diversity of human resources can be attributed to many phenomena [1], for example, globalization, the ease of movement, longer life expectancies and longer periods of professional activity, the universality of anti-discrimination regulations or increased awareness of the role of human rights and the benefits of diversity in the workplace [2,3,4,5]. Diversity management is increasingly treated as a management philosophy, which means that diversity in organizations is recognized and valued, and the goal is to increase the efficiency of the organization. This means that diversity management should be systematic and planned in the form of creating and implementing programs and procedures aimed at creating an inclusive workplace. The outlined process which organizations develop and how they implement it in the workplace should be measured. The ability to measure diversity management policies is a challenge for many organizations. Therefore, the question “How do you measure diversity and inclusion initiatives in your organization” remains valid. In addition, both practice and theory provide arguments that a process approach to diversity management ultimately affects the short-term or long-term results at the company level. The aim of this paper is to present a measurement tool that enables an assessment of the degree of development of an organization’s workplace environment in the field of diversity management. Moreover, we aim to show correlations between the individual elements of the diversity management concept architecture, including the presentation of empirical research involving 800 companies.

### Diversity, Inclusion and Diversity Management—Measurement Tools

Diversity is not a new phenomenon, rather, it has always been present in societies, which means that it has also always been present in organizations. People are different from each other, no matter how similar they are. The current visible increase in workforce diversity can be attributed to a number of factors, including, for example, globalization, ease of movement or anti-discrimination legislation and changes in the demographic structure of societies [6,7,8]. According to Drucker [9], diversity encompasses many demographic and socio-economic aspects of society, including an ageing population, an increase in workers’ competence and knowledge, increased immigration, the changing role of women in the labor market and growing cultural differences and gender roles in organizations. Classically, diversity has been defined as a mosaic of characteristics brought to the workplace environment by employees, such as gender, age, race, ethnicity, religion, family conditions and physical ability [10,11,12,13,14,15,16]. Diversity among employees can also be concerned with the diversity of functions in a given organization. It also includes the lifestyle, sexual preferences, origin and work experience at an organization, as well as the status of being employed or dismissed. Diversity, in its broad definition, can relate to any perceived difference or similarity between people, both observable and otherwise (the effect will be a wide and universal approach). Diversity is understood as a collection of characteristics, including all characteristics differentiating one person from another (in terms of employees) and their similarities.

Diversity management is a wide and complex term, and, therefore, it seems that it is difficult to set a single standard or system encompassing all aspects associated with the issue. The authors of multiple publications have defined the field and its components in different ways. An overview of them allows one to observe that diversity management is an interdisciplinary category, which utilizes several perspectives, namely, economics, social and biological ones [17,18].

Özbilgin and Tatli [19] define diversity management as a management philosophy, meaning that diversity in organizations is recognized and valued, and that the goal is to increase the performance of the organization. Egan and Bendick [20] define diversity management as a systematic and planned creation of programmes and procedures, aimed at improving the interaction between different employees (based on ethnic origin, gender and culture) in order to make diversity a source of creativity, complementarity and higher effectiveness of an organization (…). Generally, diversity management involves utilizing all available talents in an organization, without referring to ethnocentrism and stereotypes. With regards to a group of employees, it involves conducting policy on behalf of diversity, while contributing to an increase in innovativeness and creative activities, reducing any lack of human resources with regard to specified abilities and improving the quality of service for clients.

Measuring the level of diversity and inclusivity in a workplace environment is one of the most difficult parts of the process of managing these parameters. Diversity, inclusion and management are not directly observable constructs, which means they are difficult to measure [21]. Most research on diversity management has focused mainly on manager–employee interactions. Other measurement tools have focused on explaining how partial diversity management efforts affect company performance [22]. Another research approach focused on demonstrating how selected areas of organizational management (organizational climate and culture, communication and tolerance) impact on building a diverse [23,24,25,26] and inclusive organization [26,27]. It is also worth noting that measuring diversity and diversity management practices is strongly seen as a first step towards effective diversity management, which is treated as a tool to assess the starting point of an organization [28,29,30]. However, it may also be the last link in the process of implementing diversity management in an organization, as a tool used to evaluate the diversity management activities carried out to date. Continually measuring the level of implementation of diversity and inclusion measures in an organization is one of the most difficult parts of the process of managing these parameters. Diversity, inclusion and its management are theoretical constructs that are directly observable and, therefore, difficult to measure. The lack of standards for measuring diversity policy and, consequently, the lack of a universal measurement tool, have led to proposals for a number of dispersed measurement tools and their classification in the literature.

## 2. Materials and Methods

In order to achieve the aim of the study, different research methods and techniques were used, i.e., the so-called methodological triangulation was applied with regard to data sources and research methods. 

### 2.1. Description of the Research Sample

The operationalized model in the proposed simplified form, defined as the 5P architecture, underwent empirical verification in 2018 on the basis of unpublished results of a national questionnaire in a study entitled *Quantitative research on the awareness, needs and activities of companies in the field of diversity management, Lewiatan*. The study was conducted using the method of individual computer-assisted telephone interviews (CATI). The size of the basic sample from all over Poland was 800 interviews. The research was carried out on a representative random-layered sample of companies. Given the size of the population of medium and large companies (according to BDL, approx. 34,000), the average estimation error reached 3.42%. Medium and large companies located all over Poland participated in the study. The study deliberately omitted small enterprises (in smaller entities, management processes are usually so poorly developed that it is difficult to assess the adopted diversity management model—as a comprehensive approach, a process involving systematic activities and not only ad hoc activities). The respondents were representatives of the management staff or people responsible for human resource management in the organization. The characteristics of the research sample and the distribution of units within the sample are presented, among others, by the following data: female—85%, male—15%, HR specialist/manager—53%, HR director/manager—21%, board member—12%, president/vice president—6%, PR director/manager—3%, director/marketing manager—3%, owner/co-owner—2%. The analysis of the data illustrating the characteristics of the research sample allows us to conclude that the group of respondents was diverse, taking into account the following criteria: gender, age, position of the respondent and place of business or ownership. The intention was to obtain information on the awareness, needs and activities of companies in the field of diversity management. The basis of the study was a standardized questionnaire, which consisted of 33 basic questions and 7 financial record questions. The research tool developed for the purpose of the study accounted for questions that allowed the operationalization of the 5P architecture.

### 2.2. 5P Architecture

In order to assess the degree of development of an organization in the field of diversity management, an authorial proposal of five synthetic indicators was developed, corresponding to particular areas of the 5P architecture. The aim of the analysis was to enable the empirical verification of a theoretical model which combined strategic, tactical and operational levels, delineating 5 steps (areas) which should be taken to create and manage an organization open to diversity. The 5P architecture, thus, illustrates the process of implementing diversity management and its links to organizational performance. Therefore, each of the discussed areas was reviewed separately and for each of them a synthetic indicator (WP1–WP5) was created. The global diversity management indicator (hereafter W5P) was created by aggregating the sub-indicators. 

The procedure for creating each of the indicators WP1–WP5 involved analogous steps:
Indicators of diversity management in a given area were developed:
-A list of diagnostic variables was prepared by selecting appropriate questions from the questionnaire of the referenced CATI survey, with an indication of the measurement method in a given area (i.e., assigning a numerical value to individual variants of the original variable in such a way that they indicated an increasing/decreasing level of the phenomenon—the state of diversity management in a given area of P1–P5).The metric properties of an index constructed from the full set of diagnostic variables of a given 5P architecture area were investigated as follows:
(a)The internal consistency of an indicator was determined by assessing its reliability with the Cronbach’s alpha coefficient. A measurement was considered reliable if, relative to the error, it mainly reflected the true result. One of the most commonly used techniques for measuring scale reliability is the Cronbach’s alpha coefficient, which indicates to what extent a certain set of variables describes a single construct hidden in them. It can be interpreted as a measure of scale consistency (synthetic index). The coefficient can take values between 0 and 1. In this case, we estimated the proportion of the variance of the true score that was shared by the questions by comparing the sum of the variance of the questions and the variance of the total scale. If the questions did not produce a true score, but only an error (which was unknown and specific, and consequently uncorrelated across individuals), then the variance of the sum would be the same as the sum of the variances of the individual items—the coefficient alpha would be 0. If all items were perfectly reliable and measured the same thing (the true score), then the coefficient alpha would be equal to 1; values closer to 1 indicate a higher reliability of the indicator (scale). The reliability of a scale is usually considered to be high if the Cronbach’s alpha coefficient takes a value of at least 0.7. A coefficient of 0.5 is considered acceptable [31,32,33];(b)To examine the homogeneity of the index exploratory factor analysis (EFA) was used [34]. This analysis involved several steps. First, the conditions for using EFA were checked: (a) It was assessed whether the sample size was sufficient using the STV (STV ratio, n:p, i.e., the ratio between the sample size and the number of diagnostic variables). It is usually assumed that the STV should be 10:1 [35]; since the sample included 800 items and the number of diagnostic variables for each indicator was max. 39, this condition was fulfilled for each of the indicators WP1–WP5. (b) Inter-relationships between diagnostic variables were assessed using the Kaiser–Meyer–Olkin (KMO) measure of sampling adequacy and Bartlett’s sphericity test, verifying the null hypothesis that the correlation matrix between diagnostic variables was a unitary matrix [36] (if KMO > 0.5, and in Bartlett’s sphericity test gives *p* < 0.05, the set of variables is considered adequate to conduct factor analysis); (c) the diagonal values of the anti-image correlation matrix were analyzed, representing the individual values of the KMO measure for the individual variables. If the coefficients exceed 0.5, the set of variables meets the requirements of the KMO measure in relation to each item of the WP1 index separately [37]. Second, the method of extracting common variability was chosen; in each case, this was the principal components method, which is an adaptation of the classical Hotelling’s principal components method for the purposes of factor analysis and is in practice the most commonly used [38]. Thirdly, common variance resources (communality) were analyzed, i.e., a measure of the proportion of common variance presented by a given item (in other words, the amount of variance of a given item explained by factors) [39]. Fourth, in order to extract unambiguous factor structures, the Equamax orthogonal rotation was applied [40], as the correlation assessment of the distinguished components indicated a low degree of association for each of the indicators WP1–WP5. Fifth, the number of factors (components) was determined using first the Kaiser criterion (eigenvalue > 1) and then the Catell criterion (the number of factors is “cut off” when the scatter plot for the next factor becomes less steep—its slope is reduced) [41]. Sixth, the values of factor loadings, which are the correlation coefficients between a given item and the factor/component it represents, in the rotated component matrix were analyzed.
On the basis of the analyses carried out, the diagnostic variables were selected within the given NPD index for which (a) the factor loadings were greater than the adopted threshold (as recommended by Stevens [42], they should exceed 0.4), (b) the common variability resources were relatively high (arbitrarily assumed to exceed 0.1, although it should be borne in mind that the adopted threshold is liberal).Exploratory factor analysis was again conducted for the reduced set of variables and the conditions discussed in 2b were re-checked. The reliability of the new indicator was also reassessed, both overall and in possibly distinguished sub-areas.The value of the synthetic index was determined as a summary of the assessment in the selected areas (the sum of the values of the individual diagnostic variables) [33].

Finally, the distribution of the given synthetic indicators (WP1–WP5) was analyzed using basic descriptive statistics, histograms, etc. As highlighted, the diagnostic variables in each of the 5P areas were selected on the basis of substantive criteria [43].

## 3. Results

The presented approach allows the visualization of the realization of the organization’s activities in individual areas of diversity management expressed through the 5P architecture. With these individual areas (P1–P5) analyzed it is possible to assess the process in the organization in many dimensions (planning, implementation and monitoring), while at the same time giving a holistic picture of the issue under consideration. As a result of the conducted analyses, five indicators were determined which correspond to the individual stages of the organizational diversity management process.

The degree of development of diversity management in an organization can also be examined summatively for all areas of the 5P architecture. Due to the fact that some diagnostic variables are indicators of diversity management in several areas, the summary indicator W5P included 70 variables and the assessment of the reliability of the indicator was high—the Cronbach’s alpha coefficient was 0.775. The factor analysis conducted for the W5P indicator based on the set of five sub-indicators WP1–WP5 showed that within the whole process of diversity management implementation in an organization (within the 5P architecture), two sub-processes could be distinguished (Table 1). The first, which was stronger from the perspective of the organization’s real activities, was related to the operational level of diversity management (areas 2–4), while the second was related to the strategic and tactical levels (areas 1 and 5). The obtained results (high factor loadings and common variation resources, a high degree of explained variance of the latent variable—diversity management architecture 5P) confirmed the validity of the construction of the W5P index in the formula proposed in the paper, but without its disaggregation into operational, tactical and strategic levels.

### Distribution of the 5P Architecture Indicators

The WP1 index was formed as a sum of values of sub-variables, each of which could take one of two values (0 or 1), and was in the range [0, 11]. The distribution of this index is presented in Table 2.

An analysis of the distribution of individual indicators resulted in the following conclusions among medium and large companies in Poland:Diversity management in the first area (P1) was at a relatively high level. This was evidenced by averages of around 9 points against a maximum index value of 11 points. Furthermore, not only the median, but also the other quartiles reached a value of 9, which means that half of the most typical entities achieved a WP1 value of exactly 9. In fact, the distribution was tightly clustered around this value—as many as 65% of all entities surveyed achieved a 9. The differentiation of companies was small in this respect (DAR = 0.83 points); nevertheless, there were companies with an atypically low level of development in this area of diversity management. Thus, there was a strong leftward slant of the WP1 index distribution and a strong slenderness of its distribution;Diversity management in the second area (P2), i.e., implementation of specific solutions in the organization, was not as well developed. The maximum value of the WP2 index was 28 points, while within the surveyed companies the highest score was 23, and the average was a mere 5.91 (DAR = 4.60 points), which indicates the strong differentiation among companies in this area. Half of the companies achieved a WP2 of no less than 5 points, a quarter achieved no more than 2 points and the other quarter achieved no less than 9 points for this indicator. However, the skewness and flattening of this distribution were not very strong;The degree of development of diversity management in the area of P3 could be judged at a relatively high level. There was a maximum possible score of 8 points, and this value was recorded in nearly 20% of the surveyed companies, while the average was around 5.5 points (DAR = 2.18 points). Half of the companies achieved a P3 score of no less than 6, 25% no higher than 4, and for 25% this index was no lower than 7. The skewness and flattening of this distribution were weak;P4 is much less developed—with a possible maximum of 18 points, the highest WP4 value in the sample is 10, and the average is around 4.3 points (STD = 2.49 points). Half of the companies achieved a WP4 of no less than 4 points, 25% no higher than 2 points and for 25% the index is no lower than 6 points. The skewness and flattening of this distribution were weak;Like its neighbor P1, the development of diversity management in P5 was also rated highly—71% of respondents scored 14 out of a maximum of 15 points. On average, P5 scores were approximately 13.7, with very low variation (DAR = 0.96), and, thus, a strong concentration of results around the mean (K = 6.784). Not only the median, but also the other quartiles reached a value of 14. The skewness of the WP5 distribution was strong and negative (entities with a WP5 score lower than the average prevailed);The global diversity management index average was approximately 38 points, with a sample maximum of 62 points, and there was relatively little variation (DAR = 7.1 points). Half of the companies achieved a W5P of no less than 37 points, 25% no higher than 33 points, and for 25% the index was no lower than 43 points. The skewness and flattening of this distribution were weak.

The evolution of individual indicators of the 5P architecture after omitting outliers is illustrated in Figure 1. Histograms confirmed that the distribution of the W5P indicator converged to a normal distribution. Some deviations from the normal curve were visible in the case of other variables; however, taking into account their degree (relatively small), as well as the large sample size (n = approximately 800), they would not significantly affect the results of possible further analyses (alternatively, non-parametric equivalents of parametric tests can be used).

The different dimensions of the diversity management architecture interacted with each other and the direction of this correlation was positive—the development of diversity management in the different areas occurred in parallel, but, nevertheless, as can be seen from the values of the correlation coefficients, at a different pace (Table 3). Organizations that planned and programmed their diversity management vision and goals at a higher level (P1) were more likely to implement it in the subsequent stages of the 5P architecture. This translated especially into a wider range of diversity management instruments contributing to attracting and retaining a diverse workforce in the organization (P2), but also into the other elements of the 5P architecture (the least was WP3: expanding the market for goods and services).

In general, the strongest relationship with respect to the other areas of diversity management was observed for P4 and P2 (the correlation was statistically significant with respect to all other areas at *p* < 0.001, and the values of the correlation coefficients were the highest. This means that actions to promote and communicate diversity would be more extensive as the organization attracts and maintains a diverse workforce (P2), uses diversity as a competitive advantage by expanding the market for goods and services (P3), and as becomes more aware of the need to review and evaluate its diversity efforts (P5) (although this relationship was slightly weaker—rho = 0.119). Analogous relationships were also observed for the other areas. The only exception was the statistically insignificant relationship between WP3 and WP5 (rho = 0.038, *p* = 0.289)—the greater importance of market expansion as a result of diversity management implementation did not necessarily imply that there were efforts to review and evaluate diversity management.

Through a validation study with a sample of 800 companies, we tested the properties of a newly developed measure of diversity management competence and assessed the relationship between companies’ diversity management activities and perceptions of diversity-related performance and company performance. We found that activities in the area of promoting and communicating diversity were more extensive when organizations undertook more activities in the area of attracting and retaining diverse employees and used the source of diversity to increase their competitiveness through activities in the area of expanding the market for goods and services, and also when there was more awareness and activity in the area of reviewing and evaluating diversity activities undertaken in the organization. The strongest relationship among areas of diversity management was found between two of them, i.e., promoting and communicating diversity and attracting and retaining diverse employees [44]. A statistically insignificant relationship was found between expanding the market for goods and services and reviewing and evaluating the organization’s diversity efforts, i.e., increased activity in the area between expanding the market for goods and services and reviewing and evaluating efforts did not necessarily imply a strong need for openness to new markets. [45] In conclusion, the analysis of the diversity management implementation architecture in medium and large organizations operating in Poland allows us to state that the involvement of most of the surveyed companies in the analyzed subject matter was below average, and in relation to the individual components of the diversity management architecture (5P architecture), the involvement of the surveyed companies varied.

## 4. Discussion

In the study presented here, we have developed a framework and a measure of diversity management competence for the workplace environment at the company level. Our approach emphasized the process orientation of diversity management, which assumes that how a company manages diversity can ultimately affect near-term or longer-term company-level outcomes.

The proposed 5P diversity management architecture represents a novel approach to understanding how companies create a workplace environment by managing diversity from a practical point of view. The results of the development phase of our instrument have shown that company-level diversity management practices can be cost-effectively grouped into five diversity management areas. We have shown that it is possible to reliably and accurately measure these areas with the measure developed here, which demonstrated acceptable substantive, structural and criterion validity. Our study, thus, provides a practical tool to measure the workplace environment in relation to a company’s diversity management efforts that is easy to understand and “real”. Our research, therefore, reinforces the literature on the subject, showing how effective diversity management can impact company performance [46,47,48]. Our framework proposed that the intermediate outcomes of diversity management may play an intermediate role. Overall, the findings indicated that efforts to promote and communicate diversity are more extensive when organizations take more action to attract and retain a diverse workforce and to use the source of diversity to increase their competitiveness by expanding the market for goods and services, and also when there is greater awareness and activity in the area of reviewing and evaluating diversity activities undertaken in the organization. The strongest relationship among areas of diversity management was found between promoting and communicating diversity and attracting and retaining a diverse workforce. The presented approach was part of the research on how to holistically and systematically examine aspects of diversity management in an organization.

The results build on previous evidence that shows that diversity management is linked to company performance [49,50,51,52,53,54,55,56,57]. This research contributes to the literature by demonstrating how effective diversity management can affect various areas of organizational performance.

## 5. Conclusions

Our framework suggested that diversity management performance may play an indirect role affecting the workplace environment and company performance. This study suggests plausible mechanisms that may explain how diversity management can ultimately impact the workplace. It is a challenge to define these values in great depth because they are based upon human and often personal experiences—experiences that underpin people’s lives, including their lives at work, and, subsequently, their health at work [58,59]. This study provides practical guidance on managing diversity at the organizational level. First, the developed tool could be used as a diagnostic tool to assess diversity management practices in companies. The measure not only allows for diagnosis, but also suggests “good practice” in diversity management. It is also possible to use it to identify “critical areas” that can improve diversity management programs, systems and procedures. Another way to use our measure is in a normative way. A diagnostic report can be prepared that will summarize and compare (over time or between other organizations) the scope and nature of diversity management activities.

Future research directions may also be highlighted. As with many studies, there were limitations to this study. This approach represents the quantitative strand, which supports the workplace environment and organizational decision-making through the use of quantitative methods. The study’s emerging priorities are the use of quantitative data to evaluate diversity management activities and the relationships among its components. A limitation of this study was the objectivity of the qualitative data available in the study, especially in relation to the variables examined in terms of communicating and promoting diversity in the workplace environment. Nevertheless, the study certainly sets the agenda for further research. In general, today’s organizations face challenges, such as globalization, competition and changes in the labor market, and quantitative tools should play an important role in decision-making processes. Future research may include more organizations and additional industries, as well as different countries.

## Figures and Tables

**Figure 1 ijerph-19-15851-f001:**
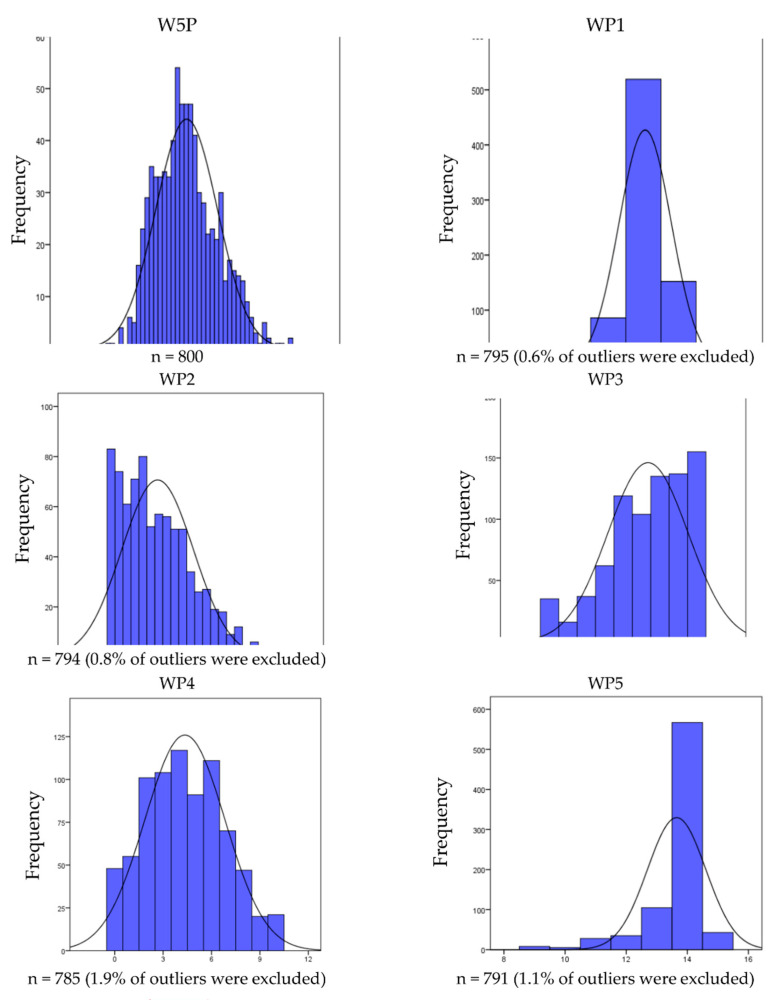
Distribution of 5P architecture indicators in medium and large companies in Poland. Source: research source.

**Table 1 ijerph-19-15851-t001:** Results of exploratory factor analysis for indicator W5P.

	Component		
	W5P.1	W5P.2	C
WP2. Recruit and retain a diverse workforce in the organization	0.706	0.056	0.501
WP3. Expanding the market for goods and services	0.681	–0.074	0.469
WP4. Promoting and communicating diversity	0.675	0.205	0.498
WP5. Review and evaluate the diversity actions taken in the organization	–0.118	0.838	0.716
WP1. Planning and programming the vision and goals of diversity management in the organization	0.266	0.741	0.619
% of explained variance for a single component	33.0	23.1	
Cumulative	33.0	56.1	
Cronbach’s alpha coefficient	0.425	0.413	

KMO = 0.604, in Bartlett’s sphericity test *p* < 0.001. Source: own research.

**Table 2 ijerph-19-15851-t002:** Basic descriptive statistics of the 5P architecture indicators.

5P Architecture Indicators	Scope	Min.	Max.	Q1	Me	Q3	M	M _ob_	STD	Ace	K
WP1. Planning and programming the vision and goals of diversity management in the organization	0 ÷ 11	4	11	9	9	9	9.02	9.08	0.83	–1.609	7.612
WP2. Recruit and retain a diverse workforce in the organization	0 ÷ 28	0	23	2	5	9	5.91	5.59	4.60	0.793	0.228
WP3. Expanding the market for goods and services	0 ÷ 8	0	8	4	6	7	5.35	5.45	2.18	–0.680	–0.235
WP4. Promoting and communicating diversity	0 ÷ 18	0	10	2	4	6	4.35	4.31	2.49	0.194	–0.628
WP5. Review and evaluate the diversity actions taken in the organization	0 ÷ 15	9	15	14	14	14	13.65	13.74	0.96	–2.355	6.784
W5P. Global Diversity Management Indicator	0 ÷ 70	20	62	33	37	43	38.09	37.89	7.10	0.425	–0.101

Q1—quartile 1, Q3—quartile 3, Me—median, M—arithmetic mean, M—trimmed mean, STD—standard deviation, As—skewness coefficient, K—kurtosis. Source: own research.

**Table 3 ijerph-19-15851-t003:** Assessment of the correlation between individual dimensions of the 5P architecture.

		WP1	WP2	WP3	WP4	WP5
WP1. Planning and programming the vision and goals of diversity management in the organization	Rho	1.000	0.360	0.158	0.258	0.270
	*p*		<0.001 **	<0.001 **	<0.001 **	<0.001 **
WP2. Recruit and retain a diverse workforce in the organization	Rho	0.360	1.000	0.227	0.325	0.072
	*p*	<0.001 **		<0.001 **	<0.001 **	0.042 *
WP3. Expanding the market for goods and services	Rho	0.158	0.227	1.000	0.243	0.038
	*p*	<0.001 **	<0.001 **		<0.001 **	0.289
WP4. Promoting and communicating diversity	Rho	0.258	0.325	0.243	1.000	0.119
	*p*	<0.001 **	<0.001 **	<0.001 **		0.001 **
WP5. Review and evaluate the diversity actions taken in the organization	Rho	0.270	0.072	0.038	0.119	1.000
	*p*	<0.001 **	0.042 *	0.289	0.001 **	

Statistically significant correlation: * α = 0.05, ** α = 0.01. Due to deviations from the assumptions made for Pearson’s linear correlation coefficient, especially the linearity of the relationship, Spearman’s rho rank correlation coefficient was used. Source: own research.

## Data Availability

Konfederacja Lewiatan, ul. Z. Cybulskiego 3, 00-727 Warszawa.

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
