# Peer review of "Towards a Better Workplace Environment—Empirical Measurement to Manage Diversity in the Workplace"

_ijerph, 2022, doi:10.3390/ijerph192315851_

Round 1
Reviewer 1 Report
The topic and aim of this research paper is atractive and resonable for current research in area of the diversity in an workplace.
However, I recommend the authors the following areas to improve and develop:
1. Quality and novelty of used references - it is necessary to add the current research papers and research results in this area, especially in the last few years the topic of diversity management is very well processing in research journals and also in business environment and practical studies (The authors used just two references of 2022 and others are more than 10 years all).
2. Research sample (deep and detail decription about research sample much be added, how the data was collected, what was the characteristics of the research sample).
3. Development of "conclusions" part - I recommed authors to focus on developing and decribing in more detail the practical and social implications of their reseach in the last part of their paperwork.
4. I recommend a minor spell check - for example try to replace verb "provide" in text by synonyms (it is overused).
Author Response
Dear Madame/Sir, thank you for your review. I tried to take into account the comments, which certainly contributed to numerous modifications of the study. I hope that the current shape of the articles meets expectations.
An additional literature review was conducted. 23 items containing the results of research from the last years (1-3 years) were attached. The research sample was clarified and described in detail.
part entitled Conclusion has been significantly modified as suggested by the reviewer.

Reviewer 2 Report
Dear authors
both content and structure of your manuscript have some significant faults and are not well enough written in terms of scientific manner:
- in abstract there is the exactly the same sentence in Methods and Results.
- again in abstract, results are somewhat described in the end, even agter conclusion
- introduction does not provide sufficient background, especially it does not argument the need for your research aim
- this is a quantitative research so hypothesis should be defined and there are none
- you say the aim is to present a measurement tool while the remaining of the manuscript tend to describe your aim as to create a model
- there are many unsubstantiated claims (for example, in lines 52, 89, 112, 125 etc.)
- only 2 references are recent, and all the others are 5 or mostly more years old
- there is a mistake in defining Cronbach's alpha in line 135, it seems like 0 means the analysis is reliable
- population and sample have not been defined, only later it is mentioned it is 800, but who, what, where...?
- there are couple of references in discussion, but not to support author's findings but to present their own results, which could be appropriate for introduction, not discussion
Author Response
Dear Madame/Sir, thank you for your review. I tried to take into account the comments, which certainly contributed to numerous modifications of the study. I hope that the current shape of the articles meets expectations.
The abstract has been revised as suggested.
The background of the topic was reformulated along with the justification for the need to achieve the goal. Unsubstantiated statements were given a source. It was clarified that the aim is to present the model that is the basis for the measurement tool of diversity management in the organization along with the conclusions resulting from the proposed model. An additional literature review was conducted. 23 items containing the results of research from the last years (1-3 years) were attached. The research sample was clarified and described in detail.
The discussion has been significantly modified as suggested by the reviewer.

Round 2
Reviewer 2 Report
Dear authors
thank you for the revised manuscript. I agree it has been significantly improved. Please conduct final spell check and try to modify some very long sentences into couple of shorter, more understandable ones (for example the last sentence of the Introduction before chapter 1.1.). Finally, in literature list in the end, please specify to which chapter of the book named as reference No 45 you referre to in the text.
Author Response
Dear Madame/Sir,
I introduced changes in the text (shortened sentences) and corrected items in the literature (45).
Elwira Gross-Gołacka
